# The role of primary care providers in testing for sexually transmitted infections in the MassHealth Medicaid program

Eric O. Mick[1], Meagan J. Sabatino[1], Matthew J. Alcusky[1], Frances E. Eanet[1], William S. Pearson[2], Arlene S. Ash[1]*

1 Department of Population and Quantitative Health Sciences, UMass Chan Medical School, Worcester, MA, United States of America, 2 Division of STD Prevention, Centers for Disease Control and Prevention, Atlanta, GA, United States of America

* arlene.ash@umassmed.edu

**Data Availability Statement:** The MassHealth (Massachusetts Medicaid and CHIP) administrative

## Abstract

The objective of this study was to determine the prevalence and predictors of testing for sexually transmitted infections (STIs) under an accountable care model of health care delivery. Data sources were claims and encounter records from the Massachusetts Medicaid and Children's Health Insurance Program (MassHealth) for enrollees aged 13 to 64 years in 2019. This cross-sectional study examines the one-year prevalence of STI testing and evaluates social determinants of health and other patient characteristics as predictors of such testing in both primary care and other settings. We identified visits with STI testing using procedure codes and primary care settings from provider code types. Among 740,417 members, 55% were female, 11% were homeless or unstably housed, and 15% had some level of disability. While the prevalence of testing in any setting was 20% (N = 151,428), only 57,215 members had testing performed in a primary care setting, resulting in an 8% prevalence of testing by primary care clinicians (PCCs). Members enrolled in a managed care organization (MCO) were significantly less likely to be tested by a primary care provider than those enrolled in accountable care organization (ACO) plans that have specific incentives for primary care practices to coordinate care. Enrollees in a Primary Care ACO had the highest rates of STI testing, both overall and by primary care providers. Massachusetts' ACO delivery systems may be able to help practices increase STI screening with explicit incentives for STI testing in primary care settings.

## Introduction

In the United States, Medicaid is a public insurance program for low-income families and individuals with disabilities; it covers more care related to sexually transmitted infections (STIs)–including syphilis, chlamydia, and gonorrhea–than any other payment source, and insures a larger proportion of the population in need of STI care [1, 2]. Care for STIs is received in various settings, with increasing national utilization trends in emergency

data used for this study included member enrollment, provider characteristics, claims, and encounter files. Such data may be requested from the Massachusetts Center for Health Information and Analysis by going to https://www.chiamass.gov/chia-data/ or directly at https://www.chiamass.gov/assets/Uploads/data-apps/Non-Government-APCD-Application.pdf. (See section VIII, Medicaid (MassHealth) Data.).

**Funding:** This publication was supported by the Centers for Disease Control and Prevention of the U.S. Department of Health and Human Services (HHS) as part of a financial assistance award totaling $113,710 with 100 percent funded by CDC/HHS. The contents are those of the author(s) and do not necessarily represent the official views of, nor an endorsement by, CDC/HHS or the U.S. Government. The funders had no role in study design, data collection and analysis, decision to publish, or preparation of the manuscript.

**Competing interests:** The authors have declared that no competing interests exist.

departments and urgent care centers [3–5]. Recent analyses of sexual health services provided in two state Medicaid programs found large differences in the location of where care was sought [6]. The American Academy of Family Physicians' "Screening for Sexually Transmitted Infections" practice manual suggests that family medicine practitioners and primary care physicians are in an ideal position to deliver routine STI screening and care to prevent transmission and future STI complications [7].

Medicaid programs are administered locally by states but regulated by the federal government through the Centers for Medicare and Medicaid Services (CMS). Massachusetts has been allowed to innovate and evaluate novel healthcare delivery modalities since 1985 [8]. The current work was conducted under the authority of the Independent Evaluation of the Massachusetts Medicaid and Children's Health Insurance Program's (i.e., MassHealth's) 1115 Demonstration Waiver Extension (2017–2022) [9].

Administrative data (claims and encounter records) tell us little about an individual's level of need for STI testing (e.g., due to the presence or absence of risky behaviors, or STI-like symptoms). Also, although we would like to learn about how organizational structures affect the comprehensiveness of STI surveillance, we cannot distinguish between tests done for diagnosis or for screening (in the absence of symptoms), nor can we find non-testing activities that have ruled out the need for testing. Thus, this study examines STI testing, an imperfect but useful proxy for STI surveillance activity.

As of 2019, over two-thirds of MassHealth's one million eligible members were enrolled in accountable care organizations (ACOs) as part of an 1115 Demonstration Waiver [10]. Through this program, MassHealth requires ACOs to engage frontline primary care practices, using value-based payments tied to cost and quality performance as incentives. In previous work analyzing MassHealth utilization, we found increased use of primary care and decreased use of acute and emergency services in the first two years following MassHealth ACO implementation in 2018. These preliminary findings suggest utilization shifts to higher-value, lower-cost care in a healthcare system that emphasizes comprehensive primary care [10–14]. However, primary care testing for STIs is neither monitored, nor specifically incentivized, within the current program, and no services to diagnose and treat STIs require referral from a members' primary care clinician (PCC).

Nonetheless, public health guidelines in Massachusetts encourage patients to consider STI-specific vaccines and to seek STI-related healthcare if they are sexually active and may have been exposed to an STI, if they are experiencing STI symptoms, or if they are considering becoming sexually active [15]. This study sought to: 1) determine the annual prevalence of STI testing both overall and specifically within primary care settings, 2) identify demographic, clinical, and social characteristics that predict STI testing, and 3) explore whether testing is more prevalent for members of ACOs than for those enrolled in other types of health plans.

## Methods

### Data

The data used for this study came from MassHealth member enrollment, provider characteristics, claims, and encounter files [14]. These records were de-identified, covered the time period from 1/1/2019 through 12/31/2019, and were analyzed in October of 2022. This study was determined to be not human subjects research by the University of Massachusetts Chan Medical School Institutional Review Board. These data belong to the Massachusetts Medicaid and Children's Health Insurance Program (MassHealth) and the research team cannot make these data publicly available themselves due to legal restrictions related to the use of data from a third party. If interested in requesting access to these data independently, researchers may

contact the Massachusetts Center for Health Information Analysis (CHIA) through the data request portal (https://www.chiamass.gov/non-government-agency-apcd-requests).

## Population

We studied managed-care-eligible members aged 13–64 years enrolled in a MassHealth managed care delivery system for at least 6 months during 2019. Children less than 13 years of age were excluded because there is little STI risk in this population. We also exclude data from the few members (contributing only about 3% of member-months) who were enrolled for less than 6 months, because they contribute too little data for determining (annual) testing prevalence. We examined 2019 data because it was the first full year of ACO program implementation in Massachusetts and was untainted by disruptions due to the COVID-19 pandemic.

## Outcomes

Our primary outcome was presence of at least one test for an STI (chlamydia, gonorrhea and/or syphilis) during 2019. We identified claims or encounters for STI testing for each member using lists of procedure, revenue, and place of service codes that identify a testing visit (using HPC codes 87491, 86631–2, 87110, 87270, 87320, 87490, 87492, 87810, 87590, 87591–2, 87801, 87850, and 86592–3; see S1 Table for code descriptions). A visit was considered to have taken place in a primary care setting if billed by a primary care provider (defined at the practice level by MassHealth), regardless of whether it was the member's assigned PCC. We identified each member as having received 1) any STI testing, and 2) any STI testing in a primary care setting.

## MassHealth healthcare delivery system

All MassHealth managed care eligible members select or are assigned a PCC, and may choose to enroll in, or accept assignment to, one of three forms of managed care: 1) a managed care organization (MCO), 2) MassHealth's PCC Case Management Plan (PCCM), or 3) an ACO. Briefly, an MCO is a type of managed healthcare plan that is intended to reduce healthcare costs by providing various economic incentives to patients and providers to select less costly forms of care and by reviewing services for medical necessity. In contrast, ACOs are groups of providers who accept responsibility for the health and costliness of a population of attributed enrollees, with ACO and provider payment linked to performance on quality metrics and meeting total cost of care benchmarks. Members enrolled in MassHealth MCOs have access to just those providers in their MCO's network. Members enrolled in the MassHealth PCCM plan have access to MassHealth's entire network of providers, with their PCC responsible for coordinating their care. There are two predominant types of ACOs: Accountable Care Partnership Plan (ACPP) ACOs and Primary Care ACOs. ACPP ACOs are integrated partnerships of a provider-led ACO and an MCO which together serve as both the health plan and the provider system for their members. Primary Care ACOs are provider-led organizations contracting directly with MassHealth (that is, without an MCO) to deliver coordinated care and manage population health [10].

## Covariates

We examined demographic, social, and clinical covariates consistent with prior studies using MassHealth administrative data, including the social determinants of health model used by MassHealth to risk adjust payments for ACOs and MCOs [16]. MassHealth uses Diagnostic Cost Group and Pharmacy Group (DxCG and RxCG) scores in program management. These scores summarize each member's total medical morbidity [17]. The DxCG model shares a

common development history with the U.S. Department of Health and Human Services' Hierarchical Condition Category (HHS-HCC) models, but is more detailed, employing a comprehensive 394-condition-category classification system [6, 18]. The DxCG model yields a relative risk score (RRS) derived from age, sex, and diagnoses recorded in clinician encounters (e.g., ambulatory care visits and hospitalizations). Specifically, we used the DxCG v4.2 concurrent model, calibrated to 2015 commercially insured data (model #88). We also used the RxCG model score (#86 in the DxCG software suite) developed on the same data. The RxCG model relies on prescriptions filled and paid for by Medicaid, rather than the diagnoses that inform the DxCG score, to summarize risk. It can infer medical problems that are not explicitly identified through recorded diagnoses, including some severity issues, such as when a member's diabetes is managed with insulin [17–19].

We identified additional variables from MassHealth claims and enrollment files: age and sex categories; disability (Medicaid entitlement due to disability); and housing problems (unstable housing, defined as having three or more addresses within the year, or ICD-10 code-identified homelessness). One further social-determinant-of-health (SDH) predictor is the Neighborhood Stress Score (NSS), calculated from seven census-block-group-level variables indicating economic stress [17, 19]. We identified a member's census block group by geocoding their most recent recorded addresses [20].

## Statistical analyses

Analyses used Stata v17.0. We used logistic regression to separately predict each of two dichotomous outcomes: any STI testing and testing in a primary care setting. Model building was influenced by our ongoing work with MassHealth. Since 2016, MassHealth has used variables like these for risk adjusting total cost of care, for primary care sub-capitation, and in quality measurement. Here we used a modified step-down model building approach, beginning by using MassHealth's most current modeling structure (i.e. age and sex, homelessness, unstable housing, disability status, the NSS7, and the DxCG and RxCG variables described above) for calculating expected total cost, and then modifying, dropping, or adding predictors based on statistical significance (p>0.05) or factors specific to the outcome of STI testing. These multivariable logistic models are used solely to establish expected levels of STI testing; odds ratios for individual variables reflect associations that should not be interpreted as causal, and some are affected by collinearity with other included variables. We measured each model's ability to predict testing using the C-Statistic (AUC or Area Under the Receiver Operating Characteristic curve) and by comparing observed rates of testing in the highest versus lowest deciles of model-predicted risk.

Using these models, we calculated the expected prevalence of testing within various subpopulations, such as members enrolled in ACOs, and compared them to observed testing rates, to identify groups receiving more or less testing than is typical for otherwise similar members of this study population. Specifically, we calculated observed-to-expected (O:E) ratios by dividing a group's actual (observed) STI testing by its model-predicted (expected) testing rates. When O:E exceeds 1.0 for a group, its members received more testing than expected, and when O:E is less than 1.0, less than expected. To test for differences in testing between plan types, we augmented the logistic regression models, adding terms for plan type. Given that we studied nearly three-quarters of a million members, with 12% in MCOs and 32% in Primary Care ACOs, the power to detect a difference in proportion between these two groups (even if it is as small as 1% and using a Type 1 error rate of alpha = 0.001) exceeds 99%.

## Results

Our study included 740,417 members. Among them 151,428 (20%) had at least some STI testing in 2019.

Table 1 shows demographic and select clinical characteristics of our study population, stratified by healthcare plan type. The full population is 55% female with 11% having some housing problems and 15% some level of disability (Table 1). Although variations in these and other tabulated variables among MassHealth plan types were modest, all variables were statistically significantly different (p<0.001) across plan types. Overall prevalence of any STI testing during CY 2019 was 20% which also varied between plans, ranging from 15% in MCOs to 22% in Primary ACO plans; testing conducted by primary care providers was much lower, being 8% overall (range 3–11%).

**Table 1. Characteristics and STI testing of 2019 MassHealth members age 13–64\*: Overall and by plan type.**

|  | ALL | | MCO | | PCCM | | ACPP | | Primary ACO | |
|---|---|---|---|---|---|---|---|---|---|---|
|  | N = 740,417 | | N = 88,721 | | N = 67,312 | | N = 346,236 | | N = 238,202 | |
|  | N | col% | N | col% | N | col% | N | col% | N | col% |
| Sex |  |  |  |  |  |  |  |  |  |  |
| Male | 333,091 | 45 | 43,487 | 49 | 29,799 | 44 | 152,986 | 44 | 106,819 | 45 |
| Female | 407,380 | 55 | 45,234 | 51 | 37,513 | 56 | 193,250 | 56 | 131,383 | 55 |
| Age Group |  |  |  |  |  |  |  |  |  |  |
| Age 13–18 years | 154,263 | 21 | 9,715 | 11 | 17,675 | 26 | 79,793 | 23 | 47,080 | 20 |
| Age 19–24 years | 82,623 | 11 | 8,671 | 10 | 7,631 | 11 | 40,433 | 12 | 25,888 | 11 |
| Age 25–44 years | 291,004 | 39 | 43,682 | 49 | 22,537 | 33 | 130,726 | 38 | 94,059 | 39 |
| Age 45–64 years | 212,581 | 29 | 26,653 | 30 | 19,469 | 29 | 95,284 | 28 | 71,175 | 30 |
| Housing Hierarchy\*\* |  |  |  |  |  |  |  |  |  |  |
| Homeless | 18,197 | 2 | 1,771 | 2 | 826 | 1 | 9,724 | 3 | 5,876 | 2 |
| Unstably housed | 66,224 | 9 | 7,094 | 8 | 6,370 | 9 | 30,846 | 9 | 21,914 | 9 |
| Neither of the above | 656,050 | 89 | 79,856 | 90 | 60,116 | 89 | 305,666 | 88 | 210,412 | 88 |
| Disability Hierarchy\*\* |  |  |  |  |  |  |  |  |  |  |
| DMH | 6,498 | 1 | 544 | 1 | 890 | 1 | 2,858 | 1 | 2,206 | 1 |
| DDS | 14,718 | 2 | 1,129 | 1 | 2,328 | 3 | 6,843 | 2 | 4,418 | 2 |
| Other disability | 91,517 | 12 | 8,030 | 9 | 10,168 | 15 | 43,703 | 13 | 29,616 | 12 |
| None of the above | 627,738 | 85 | 79,018 | 89 | 53,926 | 80 | 292,832 | 85 | 201,962 | 85 |
|  | **Mean** | **SD** | **Mean** | **SD** | **Mean** | **SD** | **Mean** | **SD** | **Mean** | **SD** |
| Continuous scores |  |  |  |  |  |  |  |  |  |  |
| Neighborhood Stress | -0.03 | 1 | -0.29 | 0.93 | -0.22 | 0.97 | 0.08 | 1.02 | -0.02 | 0.99 |
| DxCG (Model-88) | 1.27 | 2.36 | 1.18 | 2.27 | 1.3 | 2.42 | 1.26 | 2.33 | 1.31 | 2.41 |
| RxCG (Model-86) | 1.23 | 2.32 | 1.12 | 2.13 | 1.31 | 2.47 | 1.2 | 2.26 | 1.29 | 2.41 |
|  | **N** | **row%** | **N** | **row%** | **N** | **row%** | **N** | **row%** | **N** | **row%** |
| STI Testing by |  |  |  |  |  |  |  |  |  |  |
| Any Provider | 151,428 | 20.5 | 13,411 | 15.1 | 12,876 | 19.1 | 72,083 | 0.81 | 53,058 | 22.3 |
| Primary Care Provider | 57,215 | 7.7 | 2,871 | 3.2 | 7,111 | 10.6 | 19,989 | 5.8 | 27,244 | 11.4 |

\*The study population includes all managed-care-eligible members, age 13–64, and enrolled for at least 183 days in 2019.

\*\*Hierarchies are used for Housing and Disability; membership in a higher category precludes membership in a lower one.

Homeless = ICD10 code Z59.0 (Homelessness); Unstably housed = 3 or more addresses during 2019. DMH = client of the Department of Mental Health; DDS = client of the Department of Developmental Services; Other disability = Medicaid entitlement due to disability. NSS = Neighborhood Stress Score, standardized to have mean = 0 and SD = 1 in the larger MassHealth population. DxCG is Cotiviti, Inc.'s v4.2 concurrent Model 88 risk score; RxCG is its v4.2 concurrent Model 86 risk score; each is normalized to have mean = 1 in the larger MassHealth population.

**Table 2. Associations between characteristics and STI testing in 2019 MassHealth members age 13–64*.**

| | Total | Testing by Any Provider Prevalence = 20.5% | | | | | Testing by a Primary Care Provider Prevalence = 7.7% | | | | |
| --- | --- | --- | --- | --- | --- | --- | --- | --- | --- | --- | --- |
| | | C-Statistic (AUC) = 74% | | | | | C-Statistic (AUC) = 72% | | | | |
| | Total | N | % | Odds Ratio | [95% CI] | | N | % | Odds Ratio | [95% CI] | |
| Sex | | | | | | | | | | | |
| Male | 333,091 | 38,461 | 12 | Ref. | | | 13,723 | 4 | Ref. | | |
| Female | 407,380 | 112,967 | 28 | 2.82 | 2.78 | 2.86 | 43,492 | 11 | 2.58 | 2.53 | 2.63 |
| Age Group | | | | | | | | | | | |
| Age 13–18 years | 154,263 | 29,891 | 19 | 0.48 | 0.47 | 0.49 | 12,104 | 8 | 0.56 | 0.55 | 0.58 |
| Age 19–24 years | 82,623 | 29,229 | 35 | Ref. | | | 12,164 | 15 | Ref. | | |
| Age 25–44 years | 291,004 | 72,434 | 25 | 0.45 | 0.45 | 0.46 | 26,337 | 9 | 0.46 | 0.45 | 0.47 |
| Age 45–64 years | 212,581 | 19,874 | 9 | 0.12 | 0.12 | 0.13 | 6,610 | 3 | 0.14 | 0.13 | 0.14 |
| Housing Hierarchy** | | | | | | | | | | | |
| Homeless | 18,197 | 6,097 | 34 | 1.58 | 1.53 | 1.65 | 1,903 | 10 | 1.07 | 1.01 | 1.13 |
| Unstably housed | 66,224 | 19,262 | 29 | 1.29 | 1.27 | 1.32 | 7,579 | 11 | 1.23 | 1.20 | 1.26 |
| Neither of the above | 656,050 | 126,069 | 19 | Ref. | | | 47,733 | 7 | Ref. | | |
| Disability Hierarchy** | | | | | | | | | | | |
| DMH | 6,498 | 1,477 | 23 | 0.72 | 0.67 | 0.76 | 616 | 9 | 0.90 | 0.82 | 0.98 |
| DDS | 14,718 | 1,915 | 13 | 0.41 | 0.39 | 0.43 | 734 | 5 | 0.48 | 0.45 | 0.52 |
| Other Disabled | 91,517 | 14,893 | 16 | 0.72 | 0.71 | 0.74 | 5,692 | 6 | 0.85 | 0.82 | 0.87 |
| None of the Above | 627,738 | 133,143 | 21 | Ref. | | | 50,173 | 8 | Ref. | | |
| Continuous Scores | | | | | | | | | | | |
| Neighborhood Stress | | - | - | 1.19 | 1.18 | 1.20 | - | - | 1.12 | 1.11 | 1.13 |
| DxCG (per unit) | | - | - | 1.29 | 1.28 | 1.30 | - | - | 1.24 | 1.23 | 1.25 |
| DxCG (per unit, when ≥5) | | - | - | 0.67 | 0.67 | 0.68 | - | - | 0.71 | 0.70 | 0.72 |
| DxCG (per unit, when ≥20) | | - | - | 1.28 | 1.24 | 1.32 | - | - | 1.30 | 1.24 | 1.37 |
| RxCG (per unit) | | - | - | 1.18 | 1.17 | 1.19 | - | - | 1.15 | 1.13 | 1.16 |
| RxCG (per unit, when ≥5) | | - | - | 0.81 | 0.80 | 0.82 | - | - | 0.83 | 0.82 | 0.85 |
| RxCG (per unit, when≥20) | | - | - | 1.08 | 1.05 | 1.11 | - | - | 1.06 | 1.01 | 1.11 |

*The study population includes all managed-care-eligible members, age 13–64, and enrolled for at least 183 days in 2019. N = 740,471.

**Hierarchies are used for Housing and Disability; membership in a higher category precludes membership in a lower one.

Notes: Homeless = ICD10 code Z59.0 (Homelessness); Unstably housed = 3 or more addresses during 2019. DMH = client of the Department of Mental Health; DDS = client of the Department of Developmental Services; Other disability = Medicaid entitlement due to disability. NSS = Neighborhood Stress Score, standardized to have mean = 0 and SD = 1 in the larger MassHealth population. DxCG is Cotiviti, Inc.'s v4.2 concurrent Model 88 risk score; RxCG is its v4.2 concurrent Model 86 risk score; each is normalized to have mean = 1 in the larger MassHealth population.

These multivariable logistic models are used solely to establish expected levels of STI testing; odds ratios for individual variables reflect associations that should not be interpreted as causal, and some are affected by collinearity with other included variables. RxCG and DxCG scores were modeled with splines, to allow for changes in slope at values of 5 and 20.

Table 2 shows the unadjusted prevalence of STI testing (both any or in a primary care setting) within clinical and demographic characteristics, and coefficients of our multivariable models to predict the two outcomes. Patient groups with more STI testing included females (28%), young adults (35%), those with unstable housing (29%) or homelessness (34%), and those who were clients of DMH (23%). Logistic regression coefficients reflect the contribution of individual factors in the presence of other factors shown in Table 1 to predicting STI testing outcomes. Additional terms were included for both the RxCG and DxCG scores, specifically spline knots at 5 and 20, to account for the fact that both testing outcomes become increasingly

likely as these scores increase from near 0 to about 5, but less likely for those with increasingly higher levels of morbidity beyond that. The large majority (87%) of members have both RxCG and DxCG scores smaller than 5.

Our models had acceptable explanatory power (C-statistics of 74% and 72%) and could identify populations with very different levels of testing. Observed testing was 52.1% vs. 4.8% in the top vs. bottom decile of model-predicted risk for testing by any provider, and 21.2% vs. 1.5% for testing by a primary care provider. They reveal similar relationships between member characteristics and the two outcomes.

Fig 1 shows observed STI testing rates (black bars) and expected rates (grey bars) for both all testing (Fig 1A, left), and testing by primary care providers (Fig 1B), stratified by healthcare plan type: MCO, PCCM, ACPP ACO, or Primary Care ACO. It also provides O:E ratios (for example, on the left the MCO O:E ratio of 0.81 tells us that MCO testing was only 81% of what was expected based on the characteristics of MCO members, the lowest among the plan types, while the O:E ratio for Primary Care ACO members was 1.09, the highest among the plan types). Although observed rates for testing by primary care providers was more variable across plan types than all testing rates (Fig 1B), it was again true that MCO observed rates were lowest and Primary Care ACO rates highest, and that the deficit compared to expected was greatest for MCOs (O:E = 0.46) while the most "extra" testing (compared to expected) was found for Primary Care ACO members (O:E = 1.49).

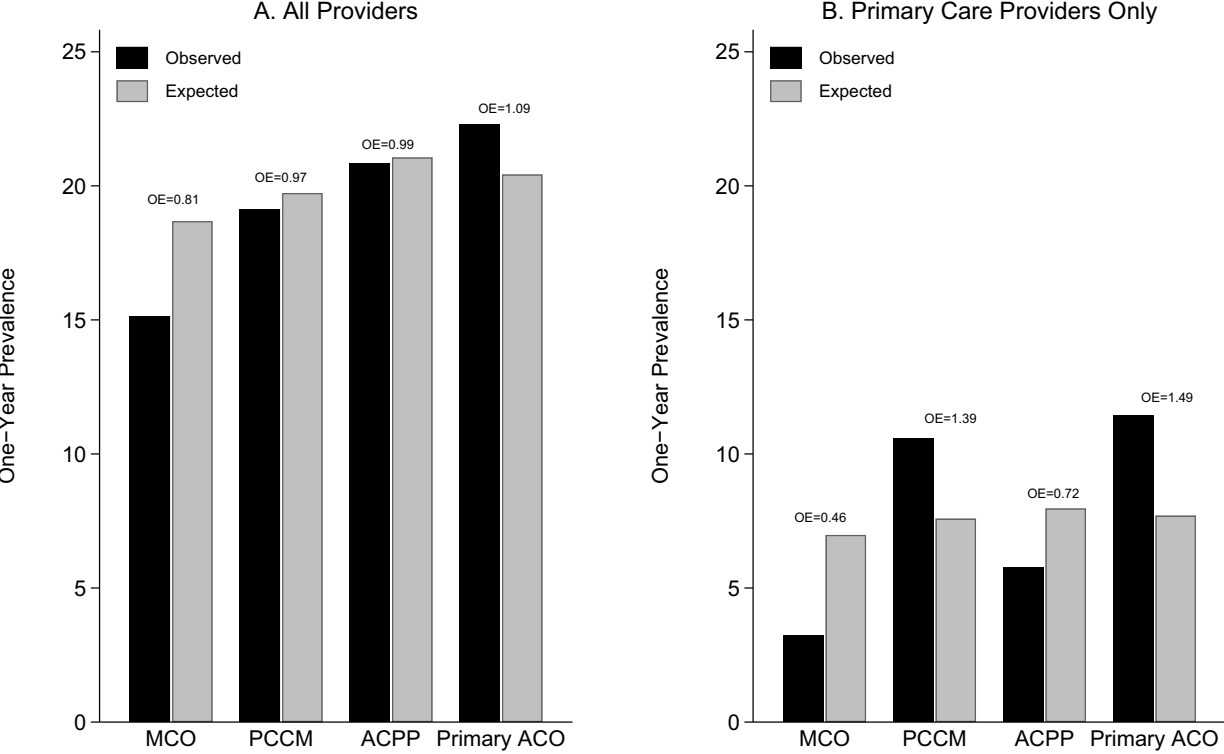

**Fig 1. Observed and risk-adjusted (expected) prevalence of STI testing in the MassHealth managed care eligible population (N = 740,471) in 2019.** Members enrolled in all healthcare plans were required to have a designated primary care provider. MCO: managed care organization (N = 88,721), PCCM: primary care case management (N = 67,312), ACPP: accountable care partnership plan (N = 346,236), Primary ACO: primary care accountable care organization (N = 238,202). Both the MCO and ACPP plans are MCO−based delivery systems where MCOs perform payment and other functions, while the PCCM and primary care ACOs are primary care−based systems that bill MassHealth directly. Expected values predicted from logistic regression models are presented in Table 2 (and S2 and S3 Tables).

## Discussion

We examined the prevalence of STI testing in the MassHealth population in 2019, the first year following MassHealth's transformation to an accountable care model of healthcare delivery. This transition focused on primary care as a way to improve integration of care across the continuum of member needs, while holding the new ACO systems and their PCCs accountable for cost and quality. We found that about one in five MassHealth members between the ages of 13 and 64 received any STI testing. Although our administrative data say little about the need for STI screening, we were able to identify demographic, clinical, and social determinants of health that could predict STI testing. We found that STI testing was more prevalent among members with housing problems, greater medical morbidity, and living in more stressed neighborhoods, but less prevalent among members with disability. Observed testing rates were about 20% less than expected for members enrolled in MCOs, but almost 10% higher than expected for members enrolled in so-called Primary Care ACOs, that are built on a foundation of primary care practices that take responsibility for population health management. These ACOs may have been more proactive than traditional MCO provider networks in identifying members for STI testing.

Studies estimate that close to one in five adults in the U.S. have a sexually transmitted infection identified throughout the year, with more than half of incident STIs occurring in persons aged 15–24 [21, 22]. In Massachusetts, the number of reported chlamydia cases increased by 38% between 2011 and 2019, reaching 31,642 reported cases, and the total number of confirmed infectious syphilis cases more than doubled during that period [23]. Gonorrhea cases reported in Massachusetts saw an approximate four-fold increase in males and nearly doubled in females during the same timeframe [23]. STI testing is recommended for Massachusetts residents who are sexually active, known to have been exposed to an STI, and those who become sexually active with a new partner [15]. Previous studies noted that Medicaid members were more likely to be identified as engaging in high-risk sexual behaviors if they were female, and between ages 15–24 and 25–34 (as compared to those aged 35–44 and 45–60) [24]. MassHealth members more likely to be tested for STIs in this study were between 19 and 24 years of age, female, had unstable housing or lived in higher stress neighborhoods, and had greater medical morbidity. Our finding that about 20% of MassHealth members aged 13–64 received STI testing is somewhat higher than findings in Medicaid populations in Maryland, where testing occurred in 16–17% and, quite a bit higher than in South Carolina, with testing in 10–11% of Medicaid members [6, 25]. Since the Maryland and South Carolina studies did not exclude members under the age of 13, the comparable prevalence calculation in Massachusetts is 14%.

MassHealth has substantially reorganized its payment and delivery systems to promote preventative care, with primary care practices at the center of the coordinated care model. Indeed, adult primary care utilization increased in early phases of the ACO program [10–12, 14]. Efforts to screen, treat, and prevent STIs are largely assumed by primary care teams, noting that education, behavioral health counseling, early diagnosis and initiation of treatment, partner notification and treatment, and vaccination are all effective STI prevention strategies [2, 26]. Routine immunizations including hepatitis A, hepatitis B, and human papillomavirus (HPV) are considered safe and efficacious for preventing some STIs, and all are currently the responsibility of primary care providers [27, 28]. Additionally, behavioral health counseling in the primary care setting has been found to reduce STI incidence in high-risk adult and sexually active adolescent populations [29]. Improved access to STI care, especially at primary care practices, is essential for screening and prevention strategies [29]. If these services are not administered by primary care providers, MassHealth members may seek care at higher cost settings, including emergency services, which are disincentivized by accountability against total cost of care benchmarks within the ACO program.

Our findings should be considered in the context of study limitations. Most important is the reliance on claims and encounter data to estimate the prevalence of STI testing, and more particularly, testing by a primary care clinical practice. While our data show where STI testing was conducted–or, more specifically, the place cited in bills–they do not reveal who ordered the test or its result. We relied on administrative files provided by the state to determine if a particular encounter was in a primary care setting. Our finding that the plans billing MassHealth directly (i.e., the PCCM plan and Primary Care ACOs) had more primary care STI testing than the MCO and ACPP ACO plans (that receive capitation payments from MassHealth and submit encounter records) may at least partially reflect how primary care activity is reported and not necessarily how it is delivered. An additional study limitation is that we do not see sexually transmitted disease care sought outside of MassHealth for reasons of confidentiality.

## Conclusions

During the transformation of the Medicaid program to an accountable care model of health-care delivery, approximately one in five MassHealth members received STI testing in 2019, with more testing among members with housing problems, moderately high levels of total medical morbidity, and living in more stressed neighborhoods. Activities directed to groups and communities with substantial social risk may be able to increase testing where it is needed most. Both before and after adjusting for differences in individual member risk, members enrolled in a Primary Care ACO had the highest rates of testing by primary care clinicians, while MCO enrollees had the lowest rates. These findings suggest that the new ACO delivery systems may hold promise for bringing practice in line with STI screening guidelines. Future work should examine ways to support primary care teams, and to reward them for providing more and better STI care.

## Supporting information

**S1 Table. Procedure codes defining the presence of testing for sexually transmitted infections.** Table includes procedure codes used to identify STI testing in the MassHealth population 13–64 years old in CY 2019.
(PDF)

**S2 Table. Logistic regression output for model testing differences between type of health care plan, adjusting for SDOH model variables, and predicting the one-year prevalence of STI testing at any provider in MassHealth managed care eligible population ages 13–64 (N = 740,471) in CY 2019.** Analyses include only those enrolled in MassHealth for at least 183 days in 2019. RxCG and DxCG scores were modeled with additional terms for splines at RRS values of 5 and 20 to model a non-linear relationship between RxCG and DxCG and the probability of having an STI-related encounter. Each unit increase is 1 standard deviation.
(PDF)

**S3 Table. Logistic regression output for model testing differences between type of health care plan, adjusting for SDOH model variables, and predicting the one-year prevalence of STI testing at a primary care setting in MassHealth managed care eligible population ages 13–64 (N = 740,471) in CY 2019.** Analyses include only those enrolled in MassHealth for at least 183 days in 2019. RxCG and DxCG scores were modeled with additional terms for splines at RRS values of 5 and 20 to model a non-linear relationship between RxCG and DxCG and the probability of having an STI-related encounter. Each unit increase is 1 standard deviation.
(PDF)

## Acknowledgments

This study was conducted under the authority of the Independent Evaluation of the 1115 Demonstration Waiver Extension (2017–2022).

## Author Contributions

**Conceptualization:** Eric O. Mick, William S. Pearson, Arlene S. Ash.

**Data curation:** Eric O. Mick, Matthew J. Alcusky.

**Formal analysis:** Eric O. Mick, Arlene S. Ash.

**Funding acquisition:** William S. Pearson.

**Methodology:** Eric O. Mick, Matthew J. Alcusky.

**Project administration:** Matthew J. Alcusky, Frances E. Eanet.

**Supervision:** Arlene S. Ash.

**Visualization:** Eric O. Mick, Meagan J. Sabatino, Matthew J. Alcusky.

**Writing – original draft:** Eric O. Mick, Meagan J. Sabatino, Matthew J. Alcusky, Frances E. Eanet, Arlene S. Ash.

**Writing – review & editing:** Eric O. Mick, Meagan J. Sabatino, Matthew J. Alcusky, Frances E. Eanet, William S. Pearson, Arlene S. Ash.

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
