## [Decision Letter · Decision Letter 0]

20 Jun 2023

PONE-D-23-14033The Role of Primary Care Providers in Testing for Sexually Transmitted Infections in the MassHealth Medicaid ProgramPLOS ONE

Dear Dr. Eanet,

Thank you for submitting your manuscript to PLOS ONE. After careful consideration, we feel that it has merit but does not fully meet PLOS ONE’s publication criteria as it currently stands. Therefore, we invite you to submit a revised version of the manuscript that addresses the points raised during the review process.

We look forward to receiving your revised manuscript.

Kind regards,

Hamufare Dumisani Dumisani Mugauri, Ph.D. Public Health

Academic Editor

PLOS ONE

Journal Requirements:

"This publication was supported by the Centers for Disease Control and Prevention of the U.S. Department of Health and Human Services (HHS) as part of a financial assistance award totaling $113,710 with 100 percent funded by CDC/HHS. The contents are those of the author(s) and do not necessarily represent the official views of, nor an endorsement by, CDC/HHS or the U.S. Government. "

"This publication was supported by the Centers for Disease Control and Prevention of the U.S. Department of Health and Human Services (HHS) as part of a financial assistance award totaling $113,710 with 100 percent funded by CDC/HHS. The contents are those of the author(s) and do not necessarily represent the official views of, nor an endorsement by, CDC/HHS or the U.S. Government. "

"This publication was supported by the Centers for Disease Control and Prevention of the U.S. Department of Health and Human Services (HHS) as part of a financial assistance award totaling $113,710 with 100 percent funded by CDC/HHS. The contents are those of the author(s) and do not necessarily represent the official views of, nor an endorsement by, CDC/HHS or the U.S. Government. "

Additional Editor Comments:

1. Principal findings: Avoid using the term "significant" in a scientific paper, unless if it denotes statistical significance

2. Conclusions: "explicit incentives for that testing incorporated into primary care settings" How sustainable is this beyond the funding duration for you to conclude effectiveness of the model based on its incentive-influenced results

3. Methodology is not explicit: study design, sampling and sample size calculation, inclusion/exclusion criteria, study period, data collection tools, ethical clearances are all missing

4. There is no evidence of how authors dealt with potential confounders or effect modifiers (table 2)

Reviewers' comments:

Reviewer's Responses to Questions

**Comments to the Author**

1. Is the manuscript technically sound, and do the data support the conclusions?

Reviewer #1: Partly

Reviewer #2: Partly

2. Has the statistical analysis been performed appropriately and rigorously? 

Reviewer #1: Yes

Reviewer #2: No

3. Have the authors made all data underlying the findings in their manuscript fully available?

Reviewer #1: No

Reviewer #2: Yes

4. Is the manuscript presented in an intelligible fashion and written in standard English?

Reviewer #1: Yes

Reviewer #2: Yes

5. Review Comments to the Author

Reviewer #1: Overall

The manuscript by Mick et al. describes prevalence of STI testing both generally and in primary care settings, within the MassHealth Medicaid programme in the United States. The background and methods were generally well written and concise. However, as someone not overly familiar with the United States system for healthcare financing, I did find some parts of the methods difficult to follow. Some brief definitions and assumption of no knowledge of such healthcare plans may help bridge the gap for international readers.

The results were interesting, particularly how STI testing in primary care was apparently enhanced by the more primary-care focussed PCCM and primary care PCO plans. However, I feel that more could be said in the discussion about the limitations of using such large datasets, without specific indicators for STI risk, particularly when assessing associations between healthcare plan and STI testing.

Background

1. “As of 2019, over two-thirds of MassHealth’s one million eligible members were enrolled in accountable care organizations (ACOs) as part of an 1115 Demonstration Waiver. Through this program, MassHealth requires ACOs to engage frontline primary care practices, using value-based payments tied to cost and quality performance as incentives.”

The authors assume reader knowledge of United States healthcare financing. To enable a more international audience to benefit from reading the paper, I suggest not assuming such knowledge. This includes, for example, including a brief definition of an “accountable care organization”, explaining what an “1115 demonstration waiver” entails, and generally providing basic descriptions of similar terms.

Methods

2. “The outcome was presence of at least one test for an STI during 2019.”

Please provide a definition of “STI” here. I note that the procedure codes for defining the presence of STI testing are in the supplementary material but think it is helpful to have in the main manuscript.

3. “MassHealth managed care eligible members may choose to enroll in, or accept assignment to, one of three forms of managed care: 1) a managed care organization (MCO), 2) MassHealth’s PCC Case Management Plan (PCCM), or 3) an ACO.”

Similarly to above, please provide definitions for MCO and PCCM.

4. Analysis strategy

There does not appear to be an explicit analysis strategy. It seems that all the variables available were put into the model. Please provide justification for this.

Results

5. “MassHealth members screened for STIs were more likely to be female, between the ages of 13 and 44, have unstable housing, live in more stressed neighborhoods, and have greater medical acuity, whether measured from diagnoses or pharmacy prescriptions.”

The age ranges are very broad, leading to a very heterogenous population within each category and making interpretation difficult. For example, 13-44 years includes both adolescents and young adults (who will likely have higher STI risk) as well as older individuals up to age 44. I note that the discussion also includes information on STI risk in different age groups (15-24, 25-34 etc) – it would be helpful to be able to compare more directly using the data in this paper.

I suggest using more narrow age categories. If these age groups are the only data available, then please state this as a limitation.

6. “MassHealth members screened for STIs were more likely to be female, between the ages of 13 and 44, have unstable housing, live in more stressed neighborhoods, and have greater medical acuity, whether measured from diagnoses or pharmacy prescriptions.”

Suggest including the unadjusted odds ratios and confidence intervals here.

7. Table 1

The totals for housing problems and disability status add up correctly in the total, no STI testing, and STI testing columns. However, those for age do not, as follows:

Total: 352,743 + 527,851 + 212,549 = 1,093,143 [stated total = 1,093,149]

No STI testing: 352,293 + 477,244 +205,941 = 1,035,478 [stated total = 940,508]

STI testing: 450 +50,607 + 6,608 = 57,665 [stated total = 152,641]

Please rectify so that numbers are correct. If missing data, please state this.

8. Table 1

I feel that row percentages would be helpful for the “no STI testing” and “STI testing” columns, rather than column percentages. For example, given that presence of at least one test for an STI is the outcome, I would argue that knowing the percentage of females (and males) who had at least one test provides more fruitful information than knowing that 50% of those who didn’t test and 74% of those who did test were female.

I acknowledge that mixing row and column percentages in a single table is not a good idea, so wonder if it might be possible to have a table purely dedicated to member characteristics, and another table showing the percentage of each variable (e.g. age group, sex) who did have an STI test.

9. Table 1

Very minor point, but I found it a bit odd to state “%” and then use decimals. Would suggest if stating percentages, to state e.g. 32% rather than 0.32. My instinct was to read it as 0.32%.

10. Table 2

I would have thought there would be a risk of collinearity including both DxCG and RxCG in the same model. Was there any evidence of this?

Discussion

11. “We found that STI testing was more prevalent among members with housing problems, greater medical morbidity, and living in more stressed neighborhoods, but less prevalent among members with disability.”

The main table that I feel is missing from the results and would be incredibly valuable would be a breakdown of characteristics (age, sex, disability status etc) between the different schemes (i.e. MCO, PCCM, ACPP, primary ACO). This would go some way to addressing one of my comments below about the differences between individuals enrolled in different schemes, and how this may affect STI testing. I appreciate these parameters are included in the models that helped you work out the observed/expected ratios, but would be helpful for the reader to see. This breakdown could be included in a new table 1 (if following my suggestion above).

12. “Primary among these is the reliance on claims and encounter data to estimate the prevalence of STI testing, and even more particularly, testing by a primary care clinical team. While our data show where STI testing was conducted – or, more specifically, the place cited in bills – it does not reveal who ordered the test or its result. We also relied on administrative files provided by the state to determine if a particular encounter was in a primary care setting. Our finding that the plans billing MassHealth directly (i.e., the PCCM plan and primary care ACOs) had more primary-care testing than the MCO and ACPP ACO plans (that receive capitation payments from MassHealth and submit encounter records) at least partially reflects technical artifacts related to how primary care activity is reported and not necessarily how it is delivered.

It is noted that plans billing MassHealth directly had more primary-care testing than the MCO and ACPP ACO plans, and how this is related to how primary care activity is reported. However, could such differences in reporting have biased the results? Could underreporting be a factor in some schemes but not others? Furthermore, are there any indicators as to the gross reliability of the underlying data?

13. There should be more discussion on the potential for confounding in influencing the differences in STI testing between different schemes, which I feel is the key limitation of this study. There is a relatively small number of variables that are adjusted for, and key variables such as presence of STI symptoms, recent contacts, or other indicators of sexual risk are not available. The variables that are included are quite general. As a result, it is difficult to really know whether it is the programmes themselves that are influencing STI testing, or whether those enrolled in different schemes have different STI risk profiles (which thus drives differences in STI testing).

I do think that it is likely that the schemes are influencing STI testing (particularly regarding STI testing in primary care for the more primary care-focussed plans). However, at present the discussion doesn’t fully address the limitations of using such data.

Reviewer #2: Why was the clinic data of infants and children used in this analysis? That is not clear. It seems the primary outcome measure was of STI testing done in "adult primary care" sites in MA for people with Medicaid. Knowing the prevalence of testing of adolescents, young adults, and older adults (13-75 for example) would be more beneficial than using the number of persons ages 0-64. Both concepts, STI screening and STI testing, are used without operational definitions. Are screening and testing equivalent concepts or is screening the sexual history taking and testing the labs ordered? There is one mention of testing for GC, CT, and syphilis in the introduction, but not in the methodology and results. Was it just GC/CT/syphilis testing or was HCV, trich, and other STI testing as well?

Also, in the discussion, I'd suggest adding something to address the higher rate of GC (and probably syphilis as well) among men in MA, but that females were more likely to be tested. Adolescents and young adults have the highest age-group incidence rates of STIs, and this is mentioned, but nothing about men who have sex with men (MSM).

I'd also bring up in the introduction and discussion something about the diagnosis and treatment of an STI at the same time testing was done. Most GC, CT, and syphilis infections are asymptomatic. Hence, routine STI testing is recommended for those with higher-risk histories (new sexual partner, 2 or more partners, substance use before sex).

Otherwise it's not clear what percent of those tested were for routine screening (asymptomatic) or not.

6. PLOS authors have the option to publish the peer review history of their article (what does this mean?). If published, this will include your full peer review and any attached files.

Reviewer #1: No

Reviewer #2: **Yes: **John A. Nelson

---

## [Author Response · Author response to Decision Letter 0]

8 Sep 2023

The reviewer and editor comments are addressed in detail in the uploaded file entitled "STI-CDC1_Response_Letter_20230908."

---

## [Decision Letter · Decision Letter 1]

24 Oct 2023

PONE-D-23-14033R1The Role of Primary Care Providers in Testing for Sexually Transmitted Infections in the MassHealth Medicaid ProgramPLOS ONE

Dear Dr. Eanet,

Thank you for submitting your manuscript to PLOS ONE. After careful consideration, we feel that it has merit but does not fully meet PLOS ONE’s publication criteria as it currently stands. Therefore, we invite you to submit a revised version of the manuscript that addresses the points raised during the review process.

We look forward to receiving your revised manuscript.

Kind regards,

Hamufare Dumisani Dumisani Mugauri, Ph.D. Public Health

Academic Editor

PLOS ONE

**Additional Editor Comments:**

Please provide all the data used in this study as required by the journal

Reviewers' comments:

Reviewer's Responses to Questions

**Comments to the Author**

1. If the authors have adequately addressed your comments raised in a previous round of review and you feel that this manuscript is now acceptable for publication, you may indicate that here to bypass the “Comments to the Author” section, enter your conflict of interest statement in the “Confidential to Editor” section, and submit your "Accept" recommendation.

Reviewer #1: (No Response)

2. Is the manuscript technically sound, and do the data support the conclusions?

Reviewer #1: Yes

3. Has the statistical analysis been performed appropriately and rigorously? 

Reviewer #1: Yes

4. Have the authors made all data underlying the findings in their manuscript fully available?

Reviewer #1: No

5. Is the manuscript presented in an intelligible fashion and written in standard English?

Reviewer #1: Yes

6. Review Comments to the Author

Reviewer #1: Thank you very much to the authors for their detailed and thoughtful responses. I only have a few additional minor suggestions for improved clarity:

1. Page 5: code 8711 should be 87110

2. “Our primary outcome was presence of at least one test for an STI during 2019. We

identified claims or encounters for STI testing for each member using lists of procedure, revenue, and place of service codes that identify a testing visit (using HPC codes 87491, 86631-2, 8711, 87270, 87320, 87490, 87492, 87810, 87590, 87591-2, 87801, 87850, and 86592-3; see Supplemental Table S1 for code descriptions).”

The definition of the primary outcome should be easily obtainable from the main paper, without referral to supplemental material. For example, “Our primary outcome was presence of at least one test for an STI (chlamydia, gonorrhoea and/or syphilis) during 2019.”

3. "Here we used a modified step-down model building approach, beginning by using MassHealth’s most current modeling structure for calculating expected total cost, and then modifying, dropping, or adding predictors based on statistical significance or factors specific to the outcome of STI testing.

a) What was the cut-off for “statistical significance” to modify, drop, or add predictors?

b) What were the starting variables in the model? (i.e. "MassHealth's most current modelling structure"

4. Table 2

The “N” column is slightly confusing, as the first “ALL” row refers to all participants, but the subsequent rows refer to the number of participants in that category who tested. Perhaps an n/N column would be more appropriate? (i.e. number of people in that category who tested for STIs/total number of people in that category) e.g. 38461/333091 for “male” testing by any provider

7. PLOS authors have the option to publish the peer review history of their article (what does this mean?). If published, this will include your full peer review and any attached files.

Reviewer #1: No

---

## [Author Response · Author response to Decision Letter 1]

7 Nov 2023

Additional Editor Comments:

Please provide all the data used in this study as required by the journal

We apologize for the confusing response to this request in the first revision. We cannot share these data ourselves as access is controlled by the state of Massachusetts, who owns the data. It is possible to request these data from the Center for Health Information Analysis and we have provided their contact information.

We now say:

“These data belong to the Massachusetts Medicaid and Children's Health Insurance Program (MassHealth) and the research team cannot make these data publicly available themselves due to legal restrictions related to the use of data from a third party. If interested in requesting access to these data independently, researchers may contact the Massachusetts Center for Health Information Analysis (CHIA) through the data request portal (https://www.chiamass.gov/non-government-agency-apcd-requests).”

Comments to the Author

6. Review Comments to the Author

Reviewer #1: Thank you very much to the authors for their detailed and thoughtful responses. I only have a few additional minor suggestions for improved clarity:

1. Page 5: code 8711 should be 87110 

Thank you this has been corrected.

2. “Our primary outcome was presence of at least one test for an STI during 2019. We

identified claims or encounters for STI testing for each member using lists of procedure, revenue, and place of service codes that identify a testing visit (using HPC codes 87491, 86631-2, 8711, 87270, 87320, 87490, 87492, 87810, 87590, 87591-2, 87801, 87850, and 86592-3; see Supplemental Table S1 for code descriptions).”

The definition of the primary outcome should be easily obtainable from the main paper, without referral to supplemental material. For example, “Our primary outcome was presence of at least one test for an STI (chlamydia, gonorrhea and/or syphilis) during 2019.”

Thank you for the suggestion which we have adopted:

Our primary outcome was presence of at least one test for an STI (chlamydia, gonorrhea and/or syphilis) during 2019. We identified claims or encounters for STI testing for each member using lists of procedure, revenue, and place of service codes that identify a testing visit (using HPC codes 87491, 86631-2, 87110, 87270, 87320, 87490, 87492, 87810, 87590, 87591-2, 87801, 87850, and 86592-3; see Supplemental Table S1 for code descriptions).

3. "Here we used a modified step-down model building approach, beginning by using MassHealth’s most current modeling structure for calculating expected total cost, and then modifying, dropping, or adding predictors based on statistical significance or factors specific to the outcome of STI testing.

a) What was the cut-off for “statistical significance” to modify, drop, or add predictors?

b) What were the starting variables in the model? (i.e. "MassHealth's most current modelling structure"

We’ve added the following to the description of the modeling:

“Here we used a modified step-down model building approach, beginning by using MassHealth’s most current modeling structure (i.e. age and sex, homelessness, unstable housing, disability status, the NSS7, and the DxCG and RxCG variables described above) for calculating expected total cost, and then modifying, dropping, or adding predictors based on statistical significance (p>0.05) or factors specific to the outcome of STI testing.”

4. Table 2

The “N” column is slightly confusing, as the first “ALL” row refers to all participants, but the subsequent rows refer to the number of participants in that category who tested. Perhaps an n/N column would be more appropriate? (i.e. number of people in that category who tested for STIs/total number of people in that category) e.g. 38461/333091 for “male” testing by any provider.

Thanks for the suggestion and we agree that it was a little confusing. Since the first ALL row only provided redundant information we’ve removed it. It was very busy with the n/N for each cell it would be required. So we added a column with totals for each subgroup to make the denominator clear.

---

## [Editor Report · Decision Letter 2]

14 Nov 2023

The Role of Primary Care Providers in Testing for Sexually Transmitted Infections in the MassHealth Medicaid Program

PONE-D-23-14033R2

Dear Dr. Eanet,

We’re pleased to inform you that your manuscript has been judged scientifically suitable for publication and will be formally accepted for publication once it meets all outstanding technical requirements.

Kind regards,

Hamufare Dumisani Dumisani Mugauri, Ph.D. Public Health

Academic Editor

PLOS ONE
---

## [Editor Report · Acceptance letter]

20 Nov 2023

PONE-D-23-14033R2 

The Role of Primary Care Providers in Testing for Sexually Transmitted Infections in the MassHealth Medicaid Program 

Dear Dr. Eanet:

I'm pleased to inform you that your manuscript has been deemed suitable for publication in PLOS ONE. Congratulations! Your manuscript is now with our production department. 

Kind regards, 

on behalf of

Mr Hamufare Dumisani Dumisani Mugauri 

Academic Editor

PLOS ONE